# Relationships Between Road and Railway Noise Exposure and Self-Reported Sleep Disturbance for Detached Houses in Japan [note 1]

**DOI:** 10.3390/ijerph22081263

**Published:** 2025-08-12

**Authors:** Makoto Morinaga, Shigenori Yokoshima, Takashi Morihara

**Affiliations:** 1School of Architecture, Department of Architecture, Daido University, Nagoya 457-0841, Japan; 2Kanagawa Environmental Research Center, Hiratsuka 254-0014, Japan; wt517310ox@kanagawa-u.ac.jp; 3Department of Architecture, National Institute of Technology, Ishikawa College, Tsubata 929-0392, Japan; morihara@ishikawa-nct.ac.jp

**Keywords:** road traffic noise, railway noise, exposure–response relationship, sleep disturbance, secondary analysis

## Abstract

This paper focuses on clarifying the relationship between noise exposure and the prevalence of highly sleep-disturbed (HSD) people due to road traffic and railway noise in Japan. The authors accumulated 22 datasets, which were provided by the Socio-Acoustic Survey Data Archive and derived from the other surveys conducted in Japan. All the datasets include the following micro-data: demographic factors, exposure, and sleep disturbance data associated with specific noise sources. We performed secondary analyses using the micro-data and established relationships between noise exposure (*L*_night_) and the percentage of HSD people (%HSD) for road traffic, conventional railway, and Shinkansen railway noise. There were no large differences in %HSD responses between noise sources, although the response to road traffic noise was slightly higher than the responses to other noise sources. In addition, the results for road traffic noise were generally consistent with those reported in the World Health Organization guidelines and previous studies conducted in Asia. In contrast, responses to railway noise, particularly in high-exposure areas, in Japan were lower than those reported in the World Health Organization guidelines and South Korean studies.

## 1. Introduction

The “Environmental Noise Guidelines for the European Region” (World Health Organization [WHO] Guidelines) [1] published in 2018 includes guideline exposure levels for various noise sources. Based on the results of a meta-analysis of surveys conducted since 2000 examining the relationship between exposure and sleep disturbance, the Guideline Development Group set guidelines for the average exposure level at night. Specifically, the guideline exposure levels for road traffic, railway, and aircraft noise were set at 45 dB, 44 dB, and 40 dB, respectively. An updated systematic review and meta-analysis by Smith et al. (2022) [2] reaffirmed these exposure–response relationships, incorporating studies up to 2021 and highlighting that transportation noise remains a significant factor in sleep disturbance across various populations. Since the publication of the WHO guidelines, discussions on revising environmental standards have been accelerating internationally. For example, according to a report by Brink (2023) [3], Switzerland has been reviewing its environmental standards and has proposed evaluating daytime noise levels using *L*_den_ and nighttime noise levels using *L*_night_. The proposed standards combine the recommendations outlined in the WHO guidelines with scientific findings from research in Switzerland.

In Japan, the maintenance of environmental quality standards (EQSs) is important for the preservation of the living environment and the protection of human health. EQSs specify separate criteria for daytime and nighttime, with *L*_Aeq_ adopted as the evaluation metric. The nighttime environmental standard is set at 40 dB in areas where quiet environments are required, while a special exception of 65 dB applies to areas adjacent to major roads. These standards primarily target road traffic noise, while there are separate environmental standards for Shinkansen railway noise and aircraft noise. However, there is no specific nighttime standard for these sources. For Shinkansen railway noise, the standard is based on *L*_Amax_ without a distinction between daytime and nighttime noise. In contrast, aircraft noise is evaluated over a 24 h period using *L*_den_. Guideline values for mitigating conventional railway noise have been established for newly constructed or significantly upgraded lines. However, no official environmental standards have been set.

As with the WHO guidelines, it is essential that noise policies, including environmental standards, are continuously reviewed based on the latest scientific knowledge in order to assess the necessity of revisions. The Institute of Noise Control Engineering/Japan (INCE/J) has established the Socio-Acoustic Survey Data Archive (SASDA), which has been operating since 2009 as a repository for social survey data related to environmental noise. Using individual data stored in SASDA, the authors previously developed a nationally representative exposure–response relationship for annoyance caused by traffic noise [4]. Although SASDA’s dataset also includes survey results on sleep disturbance and other noise-related issues beyond annoyance, these aspects were not examined.

As stated in the Night Noise Guidelines [5], sleep disturbance at night can have a negative impact on health, highlighting the need to accumulate scientific knowledge for developing noise policies addressing nighttime noise. The WHO guidelines have proposed exposure–response relationships based on a systematic review [6], including meta-analyses of the relationship between self-reported sleep disturbance and *L*_night_. Additionally, exposure–response relationships between awakening and *L*_Amax_ have been proposed based on datasets from sleep polysomnography studies, although the sample size remains limited. Because environmental standards should ideally represent long-term exposure levels, exposure–response relationships for *L*_night_ are also necessary. There have been numerous studies examining the relationship between self-reported sleep disturbance and nighttime noise exposure. For example, prior to the publication of the WHO guidelines, Miedema and Vos (2007) [7] conducted a meta-analysis based on 24 studies and developed an exposure-response relationship. Aasvang et al. (2008) [8] investigated and reported the exposure-response relationships between railway noise—specifically *L*_night_ and *L*_Amax_—and its impact on sleep. Pennig et al. (2012) [9] reported a quantitative relationship between the number of nighttime freight train noise events and awakenings. Various studies in Asian countries have also examined exposure–response relationships for sleep disturbance, particularly for road traffic noise, with research cases reported in South Korea and Hong Kong [10,11]. The findings from both studies are highly consistent, showing that at *L*_night_ = 40 dB, the prevalence of highly disturbed sleep is 1–2%, while at *L*_night_ = 50 dB, it increases to 3–4%. Railway noise studies have also been conducted in South Korea [10] and China (focusing on high-speed rail) [12]. However, unlike road traffic noise, the results of these studies do not align. For instance, at *L*_night_ = 50 dB, the South Korean study reported an 11% prevalence of highly disturbed sleep, whereas the Chinese study reported 49%. One possible reason for this discrepancy is that the Chinese study focused specifically on high-speed rail noise, which may have noise characteristics different from those of conventional railway noise.

Although many self-reported sleep-disturbance surveys have been conducted in the past, a key challenge in performing meta-analyses to provide scientific insights for noise policy is the lack of standardized survey methods for self-reported sleep disturbance. In contrast, annoyance surveys follow a standardized methodology outlined in ISO/TS 15666 [13], enabling consistent data collection. However, no such standardized approach has been established for sleep-disturbance surveys. At ICBEN 2021, the need for standardization of evaluation methods was identified as one of the key issues for future research [14]. Many datasets contained in SASDA use a 5-point scale similar to the ISO/TS 15666 method, making it relatively easier to conduct unified analyses. Although some datasets include binary yes/no questions regarding sleep disturbance, ISO/TS 15666 also suggests methods for integrating different types of response scales into a comprehensive analysis. Furthermore, Basner and McGuire (2018) [6] have reported that survey results differ depending on whether respondents are asked about sleep disturbance caused by a specific noise source or about general sleep disturbance without identifying a source. It was found that in the latter case (general sleep disturbance), the association with noise levels was not statistically significant. Consequently, the WHO guidelines adopted guidelines based on data from surveys where respondents were specifically asked about sleep disturbance caused by a particular noise source.

The aim of this study was to conduct meta-analyses of self-reported sleep disturbance caused by road and railway noise in Japan, using data deposited in SASDA as well as other datasets collected in Japan. As noted above, while studies such as those by Basner and McGuire and Miedema and Vos exist, meta-analyses on noise-induced sleep disturbance based on large-scale datasets are still relatively limited. Therefore, the meta-analysis presented in this paper using Japanese data is expected to provide valuable insights for noise policy concerning sleep-related impacts. Only data from 2000 onward are considered in order to develop a nationally representative exposure–response relationship. Specifically, the study seeks to establish exposure–response relationships between highly sleep-disturbed (HSD) people and nighttime equivalent noise levels (*L*_night_) for road traffic noise, railway noise, and Shinkansen noise, based on responses to source-specific sleep disturbance questions. Note that no datasets related to wind turbine noise are currently available in SASDA, so this source was excluded from the analysis. Similarly, only two datasets on aircraft noise are available, and in Japan, there are very few airports operating during nighttime. As a result, the number of respondents affected by nighttime aircraft noise was extremely limited, and these datasets were therefore excluded.

Regarding the impact on sleep, it is considered valuable to examine not only *L*_night_ but also the A-weighted maximum sound level (*L*_Amax_) or the influence of frequency. However, the datasets registered in SASDA do not include such information. Therefore, in this paper, we follow the approach of the WHO guidelines by using *L*_night_ for the analysis and compare the results with those from other countries. The findings are expected to provide valuable data on exposure–response relationships for sleep disturbance in Asia, where such studies remain limited. Additionally, the developed exposure–response relationships will be compared with WHO guidelines and previous studies in order to provide further discussion and insights.

## 2. Materials and Methods

### 2.1. Datasets

The 22 datasets analyzed in this paper are shown in Table 1. They were derived from 18 socio-acoustic surveys conducted from 2000 to 2023 at survey sites across Japan. Datasets 1 to 10 are deposited in SASDA, while datasets 11 to 22 were published after 2000 but are not yet registered in SASDA. It should be noted that the names of datasets 11 to 22 were assigned by the authors. Also, these dataset names are aligned with those used in the authors’ previous paper [4], which involved a meta-analysis of annoyance. Although there have been studies on aircraft noise that are not mentioned here, the number of datasets is very limited. Therefore, the analysis of aircraft noise was not included in this paper.

The sample size shows the number of data that are valid for both noise exposure and sleep disturbance rating. The sample size for each noise source is 3033, 4106, and 3644 for road traffic noise, conventional railway noise, and high-speed rail noise, respectively. As discussed in the authors’ previous paper on annoyance [4], the surveys included respondents from both detached houses and apartment buildings, leading to considerable differences in indoor exposure levels. For this analysis, which utilizes outdoor noise exposure data, it is necessary to account for differences between detached and apartment housing. However, because the majority of respondents in many surveys reside in detached houses, this analysis is performed using data from respondents living in detached houses only. Some datasets included extremely low estimated noise exposure Therefore, only respondents with estimated noise levels of 30 dB or above were selected for this analysis. Table 1 shows the sample size of respondents who satisfied these criteria.

Overall, 14 of the 22 datasets are derived from surveys conducted in areas exposed to a single specific noise. In contrast, datasets JPN021RT2004 and JPN021CR2004 are derived from a survey conducted in areas exposed to combined road traffic and conventional railway noise, while datasets KMM103CR2011 and KMM103HR2011 are derived from a survey conducted in areas exposed to both Shinkansen and conventional railway noise. Similarly, datasets KMM108CR2016 and KMM108HR2016, STM104RT2011, and STM104CR2011 are derived from a survey conducted in areas exposed to both Shinkansen and conventional railway noise. Sleep disturbance due to each noise source was obtained from one respondent in the surveys.

Between 2000 and 2006, the Ministry of the Environment conducted socio-acoustic surveys for each transportation noise source in many areas nationwide. The datasets JPN011RT2000, JPN017CR2003, and JPN018HR2003 were derived from a series of surveys. In recent years, before and after the opening of the Kyushu and the Hokuriku Shinkansen lines, thorough socio-acoustic surveys were conducted along the lines. The datasets related to the Kyushu line surveys are KMM102CR2009, KMM103CR2011, KMM103HR2011, KMM108CR2016, and KMM108HR2016 in Kumamoto prefecture. The datasets associated with the Hokuriku line surveys are NGN105HR2013 in Nagano prefecture and HKR107HR2016 in Ishikawa and Toyama prefectures. The datasets IKF109RT2019 and KNG110RT2023 were collected as part of a series of studies. The IKF109RT2019 dataset was gathered from various regions in Japan, including both urban and rural areas. The dataset KNG110RT2023 specifically focused on the Kanagawa region and involves different respondents from those in the IKF109RT2019 survey.

Except for dataset No. 22 (KNG110RT2023), the noise levels at respondents’ residences were estimated based on field measurements. Sound level measurements were conducted at multiple points near roads or railway tracks, and noise levels at residential position were then estimated using distance attenuation. In contrast, for dataset No. 22, the estimation was based on a noise map developed using a road traffic noise prediction model.

### 2.2. Evaluation of Sleep Disturbance and Number of Scale Points

Table 2 shows the number of scale points and evaluation items of sleep disturbance due to the specific noise source examined in each dataset. All the surveys evaluated self-reported sleep disturbance, which is a subjective assessment rather than an objective one. Each survey explicitly mentions the noise source when asking questions like, “Does road traffic (or railway or Shinkansen railway) noise disturb your sleep?” It has been reported that the results differ when questions about general sleep disturbances are asked without specifying the noise source [2,7]. The questionnaire items and rating scales for sleep disturbance vary across datasets, and these differences contribute to variability in the exposure–response relationship. As shown in Table 2, surveys can be divided into two types that ask about the presence of sleep disturbance and those that inquire about difficulty in falling asleep and awakenings. Surveys that ask about sleep disturbance (e.g., JPN011RT2000) for a binary response—whether the sleep disturbance is present or not, which can be interpreted as a 2-point scale. Except for the dataset IKF109RT2019, the 2-point scale surveys were designed such that respondents selected the single noise source that they felt was the most disturbing from among multiple sources and then answered whether or not their sleep was disturbed by the noise source. In the present analysis, for example, if respondents did not select railway noise as the most disturbing source in the surveys targeting railway noise, the answer was classified as having no sleep disturbance due to railway noise in the dataset. Surveys that inquire about both difficulty in falling asleep and awakenings use 5-point scales (e.g., JPN012CR2001). The 5-point scale uses descriptors similar to those used in ISO/TS 15666 for annoyance questions: “not at all,” “slightly,” “moderately,” “very,” and “extremely.” The dataset of KNG110RT2023 survey asks about frequency with the four-step options of “not at all,” “occasionally,” “1–2 times a week,” and “3 or more times a week.”

### 2.3. Noise Exposure

In this paper, nighttime noise level (*L*_night_) is defined as the equivalent continuous A-weighted equivalent sound pressure levels during nighttime (from 22:00 to 07:00). In the KNG110RT2023 survey, road traffic noise maps were created using ASJ-RTN model 2013 [15], while other surveys used measured values near the noise source to estimate exposure levels through distance attenuation. In some of the provided datasets, *L*_night_ values were not included, and only *L*_den_ values were available. In such cases, for road traffic noise, *L*_night_ was estimated from the difference between daytime and nighttime measurements at the same or similar roads. For conventional and Shinkansen railway noise, we converted *L*_den_ to *L*_night_ by calculating the ratio of train operations during daytime and nighttime based on the timetable data.

### 2.4. Method for Constructing the Exposure–Response Relationship

In annoyance studies, Schultz [16] used the percentage of highly annoyed people (%HA) to define the ratio of people who responded to either of the top two categories of a 7-point scale (cutoff point at 71%) or the top three categories of an 11-point scale (cutoff point at 73%). Meanwhile, Miedema and Vos [17] defined the upper 28% of annoyance scales (cutoff point at 72%) as %HA, assuming that the interval scale between two consecutive ratings was equidistant regardless of different modifiers of annoyance and different scale points.

In the context of these previous studies, setting the cutoff point at approximately 72% has become the de facto standard for defining high annoyance (HA). However, when the 5-point scale recommended by ISO/TS 15666 is used and respondents who answered the two top categories (4–5) are defined as HA people, the cutoff point becomes 60%. This difference in definitions of HA among datasets used in meta-analyses can introduce bias. For instance, Morinaga et al. (2021) [18] reported a roughly 5–6 dB difference in exposure–response relationships between datasets with cutoff points at 60% and 72%. Therefore, in the present analysis of sleep disturbance, the cutoff point was standardized to approximately 72%. As shown in Table 3, for a 5-point scale, respondents answering 5 are given a score of 1, and those answering 4 are given a score of 0.4, which are then summed to calculate the number of HSD responses. This approach is also discussed in ISO/TS 15666. Specifically, because each category in a 5-point scale accounts for 20%, the calculation adds the first three categories (60%) plus 60% of the 20% for the fourth category (0.2 × 0.6 = 0.12) to achieve 72%. In the case of the five-point rating scale used for questions on difficulties falling asleep and nighttime awakenings, values from Table 3 were assigned to each of the two items, and their average was used as the representative score. For example, if the rating for difficulty falling asleep was 5 and that for nighttime awakenings was 4, the resulting score would be 0.7. Similarly, for a 2-point scale, the number of responses indicating “sleep disturbance” is multiplied by 0.56 and summed. In the case of a 4-point scale, only the top category (4) is counted, resulting in a cutoff of 75%.

In the present analysis, many datasets employed multiple evaluation items for sleep disturbances, assessing both difficulty in falling asleep and nocturnal awakenings. In the systematic review [6] cited in the WHO guidelines, an average of multiple rated outcomes was calculated to determine the exposure–response relationship, and these results are adopted in the guidelines. Therefore, according to the review, when multiple items are used to assess sleep disturbance, the average value among the evaluations was used.

### 2.5. Methodology for Investigating the Effects of Gender, Age, and Changing Times

To examine the effects of demographic factors on noise-related sleep disturbance, we performed logistic regression analysis of the datasets for each transportation mode. HSD responses due to noise at a cutoff point of about 72% was set as the dependent variable, while exposure level (*NL*), gender (male and female), and age (<40, 40–59, and ≥60 years) were included as independent variables. It should be noted that *NL* was not used as a categorical variable but rather as a continuous variable. Furthermore, to examine temporal changes in sleep disturbance, the survey year was also included as a variable. Since the dataset covers approximately 20 years, from 2000 to 2023, we divided the data into two groups—those from before 2010 and those from 2010 onward—for comparative analysis.

Based on the method used in the annoyance study conducted by Miedema and Vos [17], we defined all the responses in the top category (Category 5) of the 5-point verbal scale and 40% of responses in the second-to-top category (Category 4) as highly annoyed responses by the main specific noise source. Therefore, application of the logistic regression analysis requires converting the responses in the Category 4 into HSD responses or not. In this section, we randomly divided the responses in the Category into two groups: 40% (HSD) and 60% (not HSD), following Schreckenberg’s method [19]. In a similar manner, when using a 2-point scale for the survey, 56% of respondents who answered “yes” to experiencing sleep disturbance were randomly selected to be categorized as HSD, while the remaining 44% were categorized as not HSD. Although this method is equivalent to that used by Miedema and Vos, it is also useful when performing logistic regression analysis of micro-data directly, as in the present study.

## 3. Results

### 3.1. Demographic Factors and Noise Exposure

Table 4 shows the frequency distributions of demographic factors by noise source. Values in brackets show relative frequency. There were proportionally more female respondents than male. Respondents aged 60 years or older accounted for 48–56% of the respondents for each noise source. In contrast, respondents aged less than 40 years accounted for only 11–14%.

*L*_night_ in 5 dB steps was distinguished from that in 1 dB steps, with the former abbreviated as *NL* in this paper. Table 5 displays frequency distributions of *NL* from 33 to 78 dB *L*_night_ for each transportation noise. For example, 53 dB *NL* ranges from 51 dB to 55 dB *L*_night_. Values given in parentheses represent the percentages of the corresponding items.

For road traffic noise and conventional railway noise, respondents are distributed up to *NL* = 78 dB. However, for Shinkansen noise, the upper limit is *NL* = 48 dB, with approximately 90% of respondents concentrated in the range of *NL* = 33 dB to 43 dB. It is evident that the distributions of respondents by noise level for road traffic noise and conventional railway noise are relatively similar.

### 3.2. Overview of Exposure–Sleep Disturbance Relationships

Table 6 shows the association between *NL* and %HSD for each dataset. It should be noted that Table 6 is sorted in chronological order for each noise source. In the case of sample sizes of less than 25 (hyphen in the table), we did not calculate %HSD.

Regarding road traffic noise, %HSD generally ranges from a few percent to about 20%. In areas where *NL* is 33 dB, or at 78 dB, the sample size in individual datasets is too small to calculate %HSD reliably. However, these cases are included in the meta-analysis that follows, with data from all road traffic noise studies analyzed collectively. This approach enables a more comprehensive assessment of the impact across a broader range of noise levels, even if individual datasets at specific noise levels are limited.

For conventional railway noise, %HSD typically ranges from a few percent to about 10%, indicating a slightly lower response compared with road traffic noise. However, there are cases, such as the JPN014CR2002 dataset, where %HSD exceeds 10% in areas where *NL* exceeds 63 dB. This suggests that in certain conditions or specific locations, railway noise can have a significant impact on sleep disturbance, similar to road traffic noise, particularly at higher noise levels.

For Shinkansen railway noise, *NL* peaks at 53 dB, and many data points are concentrated in areas with *NL* below 48 dB. Consequently, there are few %HSD respondents, with responses typically accounting for around 5%. This indicates that the impact of Shinkansen noise on sleep disturbance is relatively minor, likely due to the lower noise levels associated with these trains compared with conventional railways and road traffic.

One of the reasons for the wide variation in exposure levels across surveys is the differing distances from noise sources at which the social surveys were conducted. This variation derived from the fact that the locations where surveys could be implemented differed for each study. In the case of Shinkansen noise, improvements in noise mitigation measures over time may also have contributed to differences in exposure levels. Given such variability across surveys, conducting a meta-analysis that integrates these datasets allows us to examine exposure–response relationships across a broader range of exposure levels.

### 3.3. Effect of Demographic Factors and Changing Times on %HSD

Table 7 presents the results of the multiple logistic regression analysis. The area under the curve (AUC) for all noise sources was lower than 0.7. The odds ratio per 5 dB change in *NL* equals around 1.3 for all noise sources.

For all traffic noise, the odds ratio for female was below 1. While no significant differences were observed in any of the noise sources. Also, regarding age, significant differences were not obtained in all cases.

The effects of the survey period varied across noise sources. For road traffic noise, responses tended to be higher in surveys conducted after 2010, and this difference was statistically significant. Railway noise also showed a trend toward higher responses after 2010, although the difference was not statistically significant. In contrast, responses to Shinkansen noise were significantly lower in surveys conducted after 2010.

### 3.4. Establishment of Representative Exposure–Sleep Disturbance Relationship

Given that significant differences based on gender and age were not observed in all cases, exposure–response relationships were established for each noise source without considering gender and age differences. Table 8 shows the observed %HSD and sample size for each *NL* and transportation noise. In the table, hyphens indicate a sample size of <50 responses, and these were excluded when establishing the relationship.

Based on the relationship between NL and %HSD shown in Table 8, a logistic curve was estimated by applying weighting according to the number of data points at each NL level. The calculation was performed using the free statistical software R (version 4.4.3). We plotted the modeled exposure–sleep disturbance curve (solid line) and the 95% confidence interval curve (dotted line) by transportation noise, together with observed %HSD (gray circle), which are taken from the aggregated datasets and data points derived from original micro-data of each dataset in Figure 1a–c. No data point was plotted for ranges containing fewer than 25 responses. The estimated %HSD by *NL* (5 dB steps) for road traffic noise (RT), conventional railway noise (CR), and high-speed railway noise (HR) are calculated using Equations (1) to (3):(1)$$Estimated\;\%HSD\;of\;RT=1/\left\{1+EXP\left(5.6431-0.0602\times NL\right)\right\}\;(R^{2}=0.919)$$
(2)$$Estimated\;\%HSD\;of\;CR=1/\left\{1+EXP\left(5.6302-0.0528\times NL\right)\right\}\;(R^{2}=0.792)$$
(3)$$Estimated\;\%HSD\;of\;HR=1/\left\{1+EXP\left(6.3245-0.0766\times NL\right)\right\}\;(R^{2}=0.736)$$

In addition, Table 9 lists the estimated %HSD and 95% confidence interval (lower and upper) of the modeled quadratic regression. Figure 1d compares the exposure–response relationships for different noise sources, estimated at 1 dB intervals, based on Equations (1)–(3). The %HSD values for the three noise sources were generally similar, approximately 2–4% at 38 dB and around 4–7% at 48 dB. Above 48 dB, the %HSD values for road traffic noise and conventional railway noise were comparable. Although the response to road traffic noise is slightly higher than that of railway noise in the very high noise level region, the differences between both noises were not so large.

### 3.5. Comparison of Estimated Exposure–Sleep Disturabance Curves

Figure 2 compares the estimated exposure–sleep disturbance curves derived from the Japanese dataset with those derived from the WHO dataset (Guidelines) and Asian countries. Hong [10] conducted a study in South Korea on road traffic noise and railway noise, while Brown [11] investigated railway noise in Hong Kong. Both studies evaluated sleep disturbance using the 11-point scale specified in ISO/TS 15666. To define HSD individuals, both studies adopted a cutoff value of 73%, corresponding to the upper three points of the 11-point scale. The exposure–response relationships developed by WHO and this study differ in the following aspects, which should be considered when comparing the two:-The WHO exposure–response relationship does not necessarily use a cutoff point of 72% for defining %HSD; some datasets include cutoff values set at 60% or 67%.-Although WHO calculates sleep disturbance by averaging scores for multiple indicators, such as nocturnal awakenings and difficulty falling asleep, this study adopts the maximum value of these scores.

For road traffic noise, the overall trends in all four studies are similar. However, the response in Japan was slightly higher than those of other studies, although the differences are small. For example, at *L*_night_ = 40 dB, the %HSD is 5% in the present study, while the WHO results and Hong [10] are both 2%, and Brown [11] reports 1%. At *L*_night_ = 50 dB, the %HSD is 8% in this study, 4% in both WHO and Hong [10], and 3% in Brown [11]. These findings indicate a general consistency across studies. Regarding railway noise, high-speed trains such as the Shinkansen are not covered by the WHO guidelines; thus, the results of Shinkansen noise in the present study are provided for reference only. For both conventional and high-speed railways, the exposure–response relationship in this study is mostly consistent with the WHO curve for *L*_night_ ≤ 48 dB. At levels exceeding 48 dB, the WHO’s response tends to be higher. For example, at *L*_night_ = 48 dB, the present study’s CR and the WHO Guidelines have a %HSD of 4% and 5%, respectively. At *L*_night_ = 60 dB, the present study’s CR is 8%, while that of the WHO is 17%. The exposure–response relationship for railway noise shown in Hong [10] is higher than other three curves.

## 4. Discussion

The results of this study, which established exposure–response relationships for self-reported sleep disturbance caused by road traffic noise, conventional railway noise, and Shinkansen railway noise, showed no large differences between noise sources, as illustrated in Figure 1d. In contrast, previous studies analyzing annoyance using the SASDA dataset and social survey data from Japan found significant differences in exposure–response relationships depending on the noise source [4]. Even at the same *L*_den_, annoyance responses were highest for military aircraft noise, followed by civil aircraft noise, Shinkansen noise, conventional railway noise, and road traffic noise. For example, at *L*_den_ = 55 dB, the percentage of highly annoyed people was approximately 10% for road traffic noise, 20% for conventional railway noise, and 30% for Shinkansen noise. These findings suggest that annoyance responses are strongly influenced by non-acoustic factors such as attitudes toward the noise source. “Attitude toward noise” refers to one’s perception of the noise source, shaped by factors such as the perceived social benefits of the transportation system causing the noise and the personal advantages one may gain from it [20]. In contrast, social responses to sleep disturbance, even when self-reported, appear to be more directly related to acoustic factors. Tetsuya et al. (2017) [21] have previously investigated the effects of step changes in noise exposure on social responses and found that annoyance responses increase disproportionately when noise exposure levels rise sharply. However, for self-reported sleep disturbance, no excessive reaction was observed in response to abrupt noise exposure changes [22]. A review of intervention studies by Brown and van Kamp [23] reported similar findings. The fact that no large differences in sleep disturbance responses were observed across noise sources in the present meta-analysis aligns with the insights from these previous studies.

Examining the logistic regression analysis results in Table 7, the AUC values for the models of all noise sources fall below 0.7, which is generally considered a threshold for a well-fitted model. This suggests that these models do not provide a strong fit to the data. In particular, the model fit for Shinkansen noise is notably poor. One key factor contributing to this result is the limited nighttime operation of Shinkansen trains, which generally run between 6:00 AM and 11:30 PM, depending on the area. As a result, opportunities for people to be exposed to Shinkansen noise during typical sleeping hours are inherently limited compared to other noise sources. It should be noted that this does not imply that the impact of Shinkansen noise on sleep is small, but rather that the chance of being exposed to such noise during the defined nighttime period is lower. In effect, the evaluation is limited to individuals who sleep between 10:00 PM and 11:30 PM, and between 6:00 AM and 7:00 AM—periods when Shinkansen operations coincide with the nighttime hours. Additionally, as reported by Yokoshima et al. [24], responses to Shinkansen and railway noise often include the influence of vibrations. This may further explain the low model fit, given that the current model considers only noise levels and does not account for vibration effects. In contrast, the road traffic and railway noise models demonstrated a moderately fit, with a clear exposure–response relationship. In Japan, many areas along major roads are subject to special exemptions in EQSs, with the nighttime *L*_Aeq_ set at 65 dB. As a result, a significant portion of the population is exposed to high nighttime noise levels., Because trains operate past 10 PM and, in urban areas, until around midnight, a clearer exposure–response relationship was observed for railway noise compared with Shinkansen noise.

The results of the logistic regression analysis examining differences in responses before and after 2010 revealed varying trends depending on the noise source. For road traffic noise, responses were higher in surveys conducted after 2010, whereas the opposite trend was observed for Shinkansen noise. Regarding road traffic noise, Figure 1a shows that the higher responses in the 2019 survey (IKF109RT2019) and the 2023 survey (KNG110RT2023) appear to be the main contributors to this increase. On the other hand, the 2011 survey (STM109RT2011) showed the lowest response, suggesting that a simple interpretation of increased responses over time is not straightforward. It has been reported that the response rates in these 2 surveys were relatively low, around 20%, which may be related to the high levels of responses. In contrast, responses to Shinkansen noise have significantly decreased in recent years. Recent social surveys on Shinkansen noise have been conducted for newly opened lines such as the Hokuriku and Nagano Shinkansen. These newer Shinkansen lines have incorporated improved vibration mitigation measures, which may have contributed to the lower responses.

As shown in Figure 2, the results for road traffic noise align closely with the findings from all other studies. However, for conventional railway noise, particularly at levels above 50 dB, the WHO guidelines and Hong et al.’s study in South Korea reported higher responses compared with the present study. One possible explanation is that the WHO guidelines include numerous studies conducted in regions where nighttime freight train operations contribute significantly to noise exposure [6], which may have led to higher rates of reported sleep disturbance. Hong [10] compared its results with the exposure–response relationships established by Miedema and Vos [17], which were primarily based on European data. The study found that South Korean responses were higher and suggested several possible reasons, including closer proximity to railway tracks, differences in background noise levels, and differences in annoyance sensitivity toward railway noise.

However, given that no quantitative data were provided to support these claims, a direct comparison remains challenging. As mentioned above, if we assume that non-acoustic factors have a relatively smaller influence on sleep disturbance compared with annoyance, it is possible that acoustic factors—including differences in vibration impacts due to distance from railway tracks or diffraction attenuation caused by surrounding buildings—may have contributed to these variations. Notably, as shown in Figure 2b, the results from this study are relatively similar to the meta-analysis performed by Miedema et al., which focused primarily on European data and was also used as a comparison reference in Hong [10].

One limitation of this study is that it does not take into account individual factors related to noise, such as noise sensitivity and the vulnerability of certain groups, including children. Although we consider the influence of non-acoustic factors (e.g., attitudes toward noise sources) to be relatively small, we did not explore physiological mechanisms related to individual characteristics. Previous studies have suggested that noise sensitivity affects sleep disturbance. Li et al. [25] reported that individuals with high noise sensitivity experience greater sleep disturbance. Additionally, Halonen et al. [26] showed that *L*_night_ was associated with insomnia symptoms among persons with higher scores for trait anxiety. Regarding noise sensitivity, future issues include defining sensitivity, standardizing assessment methods, and elucidating mechanisms linking noise sensitivity to noise-induced effects. Addressing these issues will be essential for improving the accuracy and applicability of research on individual differences in noise susceptibility. Regarding children, Weyde [27] reported that girls are particularly susceptible to sleep disturbances caused by noise, although scientific evidence on this issue remains limited. Given that children are considered a vulnerable group in terms of noise exposure, it is crucial to accumulate more research data on the effects of noise on children in future studies.

Also, when examining the exposure–response relationship based on outdoor noise levels, it is essential to consider the sound insulation performance associated with building structures and window types. In this study, we limited the analysis to detached houses, as apartment buildings generally have better sound insulation than detached houses. However, even among detached houses, the difference between indoor and outdoor sound pressure levels can vary depending on construction types. Furthermore, there are cases in which respondents sleep with their windows open. Since these factors could not be taken into account in the present analysis, this represents one of the limitations of the study.

In Japan, noise policies based on noise maps—such as those commonly implemented in Europe—are still extremely limited, and the dissemination of noise maps has been slow. As a result, social surveys utilizing noise maps have rarely been conducted. Consequently, the present analysis relies on exposure estimates based on measurements conducted at limited locations, leaving issues related to uncertainty in exposure levels. Recently, however, surveys such as dataset No. 22 have begun to incorporate noise maps, and in the future, the wider implementation of noise maps is expected to enable the development of more accurate exposure–response relationships.

## 5. Conclusions

This paper established the representative relationship between noise exposure and the prevalence of self-reported sleep disturbance due to road traffic and railway noise in Japan. We accumulated 22 datasets, which were deposited in SASDA or derived from the other surveys conducted in Japan. All the datasets include the following micro-data: demographic factors (age and gender) as well as exposure and sleep disturbance data associated with specific transportation noise. Using micro-data, we performed a secondary analysis and established a modeled exposure-sleep disturbance relationship by noise source.

The results of this study, which established exposure–response relationships for road traffic noise, conventional railway noise, and Shinkansen railway noise, showed that there were no large differences in responses among noise sources. However, responses to road traffic noise were slightly greater than those for conventional railway noise and Shinkansen railway noise, even though no data were available for Shinkansen noise above 48 dB. Additionally, for road traffic noise, the results were generally consistent with those reported in the WHO guidelines as well as previous studies conducted in Asia. In contrast, for railway noise—particularly in high-exposure areas—responses in Japan were lower than those reported in the WHO guidelines and South Korean studies. Possible contributing factors, such as vibration intensity and distance from railway tracks, were considered in the discussion.

Finally, in Japan, whether the WHO guideline recommendation values can be applied to the standard value for EQSs for the noise sources or not is currently under discussion. The EQSs for environmental noise are indispensable for planning noise policy and creating effective countermeasures. The EQSs are established or revised whenever necessary based on the new scientific knowledge of environmental noise effects on humans obtained in this study. For self-reported sleep disturbance, the results from Japan and the WHO guidelines were relatively similar. However, some discrepancies were observed, particularly in areas with high exposure to railway noise. Additionally, there is a lack of sufficient datasets to establish exposure–response relationships for aircraft noise, highlighting the need for further data collection in this area. The application of the findings in this study is limited because the results are only based on the sleep disturbance due to road traffic and railway noise in Japanese detached houses. Therefore, data on sleep disturbances in apartments, as well as aircraft noise data, are also needed.

## Figures and Tables

**Figure 1 ijerph-22-01263-f001:**
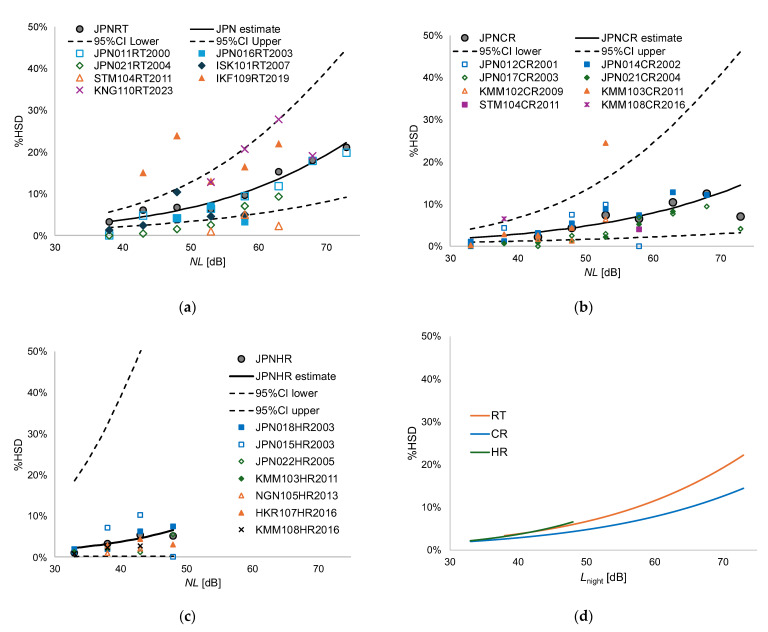
Estimated exposure–sleep disturbance relationships (line) taken from the aggregated data and data points derived from micro-data: (**a**) road traffic noise; (**b**) conventional railway noise; (**c**) Shinkansen railway noise; (**d**) comparison among noise sources. Subfigure a–c was plotted in 5 dB units of the noise level. Subfigure d was drawn by interpolating the noise level in 1 dB units. “CI” means confidence interval.

**Figure 2 ijerph-22-01263-f002:**
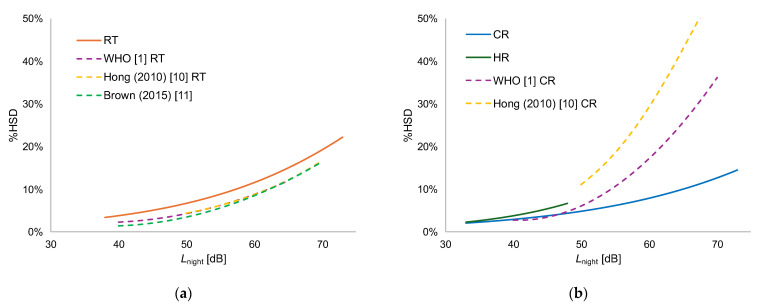
Comparison between estimated exposure–annoyance curves derived from Japanese, Asian, and WHO datasets: (**a**) Japanese road traffic noise (JPN-RT) is compared with WHO road traffic noise (WHO [1] RT), South Korea road traffic noise (Hong (2018) [10] RT), and Hong Kong toad traffic noise (Brown (2015) [11]); (**b**) Japanese conventional railway noise (JPN-CR) and Shinkansen railway noise of (JPN-HR) are compared with WHO railway noise (WHO [1] CR), and South Korea train noise (Hong (2018) [10] CR). Both figures were drawn by interpolating the noise level in 1 dB units.

**Table 1 ijerph-22-01263-t001:** Outline of datasets analyzed in this paper. CR, conventional railway noise; HR, high-speed railway noise; RT, road traffic noise.

No	Dataset	Source	Survey Year	Survey Site	Sample Size
1	JPN011RT2000	RT	2000–2006	Saitama, Chiba, Tokyo, Kanagawa, Nagano, Osaka, and Fukuoka Prefs.	1028
2	JPN012CR2001	CR	2000	Hokkaido Pref.	394
3	JPN014CR2002	CR	2002	Fukuoka Pref.	1483
4	JPN015HR2003	HR	2003	Fukuoka Pref.	683
5	JPN016RT2003	RT	2003–2004	Hokkaido Pref.	272
6	JPN017CR2003	CR	2003–2006	Chiba, Tokyo, Kanagawa, Aichi, Osaka, and Kumamoto Prefs.	712
7	JPN018HR2003	HR	2003–2006	Tochigi, Saitama, Tokyo, Kanagawa, Nagano, Shizuoka, and Osaka Prefs.	824
8	JPN021RT2004	RT	2004–2006	Kanagawa Pref.	630
9	JPN021CR2004	CR	2004–2006	Kanagawa Pref.	609
10	JPN022HR2005	HR	2005	Nagoya City	110
11	ISK101RT2007	RT	2007	Ishikawa Pref.	353
12	KMM102CR2009	CR	2009–2010	Kumamoto Pref.	242
13	KMM103CR2011	CR	2011–2012	Kumamoto Pref.	431
14	KMM103HR2011	HR	2011–2012	Kumamoto Pref.	467
15	STM104RT2011	RT	2011	Saitama City	131
16	STM104CR2011	CR	2011	Saitama City	107
17	NGN105HR2013	HR	2013	Nagano Pref.	224
18	HKR107HR2016	HR	2016	Ishikawa and Toyama Prefs.	914
19	KMM108CR2016	CR	2016	Kumamoto Pref.	128
20	KMM108HR2016	HR	2016	Kumamoto Pref.	422
21	IKF109RT2019	RT	2019	Ibaraki, Kanagawa and Fukuoka Prefs.	359
22	KNG110RT2023	RT	2023	Kanagawa Pref.	260

**Table 2 ijerph-22-01263-t002:** List of the number of scale points and evaluation items of sleep disturbance due to a specific noise source.

No.	Dataset	Scale Point	Evaluation Items
1	JPN011RT2000	2	Sleep disturbance
2	JPN012CR2001	5	Difficulty in falling asleep + awakening
3	JPN014CR2002	5	Difficulty in falling asleep + awakening
4	JPN015HR2003	5	Difficulty in falling asleep + awakening
5	JPN016RT2003	5	Difficulty in falling asleep + awakening
6	JPN017CR2003	2	Sleep disturbance
7	JPN018HR2003	2	Sleep disturbance
8	JPN021RT2004	5	Difficulty in falling asleep + awakening
9	JPN021CR2004	5	Difficulty in falling asleep + awakening
10	JPN022HR2005	2	Sleep disturbance
11	ISK101RT2007	5	Difficulty in falling asleep + awakening
12	KMM102CR2009	5	Difficulty in falling asleep + awakening
13	KMM103CR2011	5	Difficulty in falling asleep + awakening
14	KMM103HR2011	5	Difficulty in falling asleep + awakening
15	STM104RT2011	5	Difficulty in falling asleep + awakening
16	STM104CR2011	5	Difficulty in falling asleep + awakening
17	NGN105HR2013	5	Difficulty in falling asleep + awakening
18	HKR107HR2016	5	Difficulty in falling asleep + awakening
19	KMM108CR2016	5	Difficulty in falling asleep + awakening
20	KMM108HR2016	5	Difficulty in falling asleep + awakening
21	IKF109RT2019	2	Sleep disturbance
22	KNG110RT2023	4	Difficulty in falling asleep + nocturnal awaking + early-morning awakening + feeling do not sleep well + daytime sleepiness

**Table 3 ijerph-22-01263-t003:** Converted score for each point scale.

Number of Scale Points	Category				
	1	2	3	4	5
2-point scale	0	0.56	-	-	-
4-point scale	0	0	0	1	-
5-point scale	0	0	0	0.4	1

**Table 4 ijerph-22-01263-t004:** Frequency of distributions of demographic factors for each noise source. CR, conventional railway noise; HR, high-speed railway noise; RT, road traffic noise.

Item	Gender			Age (Years)			
Noise	Male	Female	No Answer	<40	41–59	60≤	No Answer
RT	1399 (46%)	1576 (52%)	58 (2%)	424 (14%)	1021 (34%)	1523 (50%)	65 (2%)
CR	1745 (42%)	2310 (56%)	51 (2%)	578 (14%)	1527 (37%)	1964 (48%)	37 (1%)
HR	1677 (46%)	1926 (53%)	41 (1%)	391 (11%)	1180 (32%)	2044 (56%)	29 (1%)

**Table 5 ijerph-22-01263-t005:** Frequency distributions of noise exposure for each noise source. *NL*, *L*_night_; CR, conventional railway noise; HR, high-speed railway noise; RT, road traffic noise.

	*NL* (dB)	33	38	43	48	53	58	63	68	73	78
Noise Source											
RT		27 (1%)	187 (6%)	373 (12%)	621 (20%)	707 (23%)	517 (17%)	330 (11%)	183 (6%)	85 (3%)	3 (0%)
CR		335 (8%)	406 (10%)	693 (17%)	803 (20%)	706 (17%)	579 (14%)	363 (9%)	154 (4%)	58 (1%)	8 (0%)
HR		570 (16%)	1514 (42%)	1196 (33%)	306 (8%)	-	-	-	-	-	-

**Table 6 ijerph-22-01263-t006:** %HSD as a function of *NL* (5 dB step in *L*_night_) for each dataset.

Dataset	*NL* (dB)								
	33	38	43	48	53	58	63	68	73
JPN011RT2000	-	0	5	4	6	10	12	18	20
JPN012CR2001	-	4	3	8	10	-	-	-	-
JPN014CR2002	1	1	3	5	9	7	13	12	-
JPN015HR2003	1	7	10	-	-	-	-	-	-
JPN016RT2003	-	-	-	4	7	3	-	-	-
JPN017CR2003	-	-	0	3	3	7	8	9	4
JPN018HR2003	2	2	6	7	5	-	-	-	-
JPN021RT2004	-	0	1	2	3	7	9	-	-
JPN021CR2004	0	1	1	1	2	5	8	-	-
JPN022HR2005	-	-	1	5	-	-	-	-	-
ISK101RT2007	-	1	2	10	5	5	-	-	-
KMM102CR2009	-	-	2	1	6	-	-	-	-
KMM103CR2011	0	3	3	4	24	-	-	-	-
KMM103HR2011	1	2	-	-	-	-	-	-	-
STM104RT2011	-	-	-	-	1	5	2	-	-
STM104CR2011	-	-	-	-	-	4	-	-	-
NGN105HR2013	-	1	2	-	-	-	-	-	-
HKR107HR2016	-	3	4	3	-	-	-	-	-
KMM108CR2016	0	6	-	-	-	-	-	-	-
KMM108HR2016	1	2	3	-	-	-	-	-	-
IKF109RT2019	-	-	15	24	13	16	22	-	-
KNG110RT2023	-	-	-	-	13	21	28	19	-

**Table 7 ijerph-22-01263-t007:** Exposure–annoyance relationships for each dataset. Odds ratio reflects a change of 1 dB in the noise level. CR, conventional railway noise; HR, high-speed railway noise; RT, road traffic noise; *NL*, *L*_night_; AUC, area under the curve; O.R., odds ratio based on the following reference category: male (gender), and −40 (age); S.E., standard error; 95% LCI and UCI stand for lower and upper limits of a 95% confidence interval, respectively. An asterisk displayed as a superscript next to a variable name in the item column indicates that the variable is statistically significant at the 5% level.

	Item	Category	Estimate	S.E.	*p*	O.R.	95% LCI	95% UCI
RT	*NL* *		0.052	0.008	0.000	1.054	1.038	1.069
*n* = 3033	Gender	Female	–0.061	0.134	0.652	0.941	0.723	1.225
AUC = 0.68	Age	40−59	0.093	0.211	0.659	1.098	0.732	1.687
		≥60	–0.090	0.205	0.662	0.914	0.617	1.385
	After 2010 *		0.873	0.136	0.000	2.395	1.831	3.126
	Constant		−5.395	0.469	0.000	0.005	-	-
CR	*NL* *		0.062	0.008	0.000	1.064	1.047	1.081
*n* = 4106	Gender	Female	−0.201	0.150	0.181	0.818	0.610	1.099
AUC = 0.67	Age	40−59	0.189	0.236	0.422	1.208	0.772	1.956
		≥60	−0.090	0.236	0.704	0.914	0.584	1.480
	After 2010		0.258	0.211	0.222	1.295	0.843	1.936
	Constant		−6.176	0.496	0.000	0.002	-	-
HR	*NL* *		0.059	0.017	0.001	1.061	1.025	1.097
*n* = 3644	Gender	Female	–0.125	0.172	0.468	0.882	0.629	1.239
AUC = 0.64	Age	40−59	−0.289	0.285	0.310	0.749	0.435	1.336
		≥60	−0.046	0.264	0.861	0.955	0.582	1.645
	After 2010 *		−0.654	0.179	0.000	0.520	0.364	0.737
	Constant		−5.075	0.759	0.000	0.006	-	-

**Table 8 ijerph-22-01263-t008:** Observed %HSD and sample size of *NL* by transportation noise. *NL*, *L*_night_; CR, conventional railway noise; HR, high-speed railway noise; RT, road traffic noise. Values in each cell are as follows: top, %HSD; bottom, sample size.

Source	*NL* (dB)								
	33	38	43	48	53	58	63	68	73
RT	-	3 187	6 373	7 621	6 707	10 517	15 330	18 183	21 85
CR	0 335	2 406	2 693	4 803	7 706	7 579	10 363	12 154	7 58
HR	1 570	3 1514	5 1196	5 306	-	-	-	-	-

**Table 9 ijerph-22-01263-t009:** Estimated %HSD and 95% confidence interval of logistic curve for 5 dB steps of *NL*. *NL*, *L*_night_; CR, conventional railway noise; HR, high-speed railway noise; RT, road traffic noise; *NL*, *L*_night_; %HSD, percent of highly sleep-disturbed people; 95%CI, 95% confidence interval.

*NL*	RT		CR		HR	
	%HSD	95%CI	%HSD	95%CI	%HSD	95%CI
33	-	-	2.0	1.0–4.1	2.2	0.2–18.6
38	3.4	2.0–5.6	2.6	1.1–5.8	3.2	0.2–32.2
43	4.5	2.5–8.0	3.4	1.3–8.3	4.6	0.2–49.8
48	6.0	3.1–11.2	4.3	1.5–11.6	6.6	0.6–67.4
53	7.9	3.9–15.4	5.6	1.8–16.0	-	-
58	10.4	4.8–20.9	7.1	2.1–21.8	-	-
63	13.5	6.0–27.7	9.1	2.4–28.8	-	-
68	17.5	7.5–35.7	11.5	2.8–37.1	-	-
73	22.2	9.2–44.6	14.5	3.2–46.1	-	-

## Data Availability

This study re-analyzes Socio-Acoustic Survey Data Archive (SASDA), which are available at https://www.ince-j.or.jp/old/04/04_page/04_doc/bunkakai/shachodata/?page_id=972 (accessed on 20 May 2025), as well as other datasets not stored in SASDA, for which we obtained consent for secondary analysis from the project manager of each survey.

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
