# Peer review of "Relationships Between Road and Railway Noise Exposure and Self-Reported Sleep Disturbance for Detached Houses in Japanâ€"

_ijerph, 2025, doi:10.3390/ijerph22081263_

Round 1
Reviewer 1 Report
Comments and Suggestions for Authors
This manuscript presents a valuable analysis of exposure-response relationships between road traffic, conventional railway, high-speed railway noise and sleep disturbance in Japan, leveraging a robust dataset from SASDA and other surveys. The study addresses an important gap in regional noise policy development and provides meaningful comparisons with WHO guidelines and Asian studies. The study's approach to processing historical data largely adhered to international standards, and the research findings align closely with similar studies, demonstrating strong reliability. However, there are still some problems that authors should notice, the details are listed as follow:
- The manuscript aggregates datasets spanning 23 years (2000–2023), which introduces potential temporal biases. Why do not address generational shifts in noise sensitivity or lifestyle changes over this period? Please conduct intergenerational comparative analyses to examine whether societal changes or cumulative noise pollution exposure influence subjective sleep disorder reporting.
- The harmonization of sleep disturbance metrics across datasets raises concerns. Surveys used divergent scales: binary (yes/no) or 5-point Likert scales evaluating difficulty falling asleep and awakenings. Why the manuscript combine responses by taking the maximum value between difficulty falling asleep and awakenings? This approach assumes equivalence between these distinct phenomena, though they may reflect different physiological and psychological impacts. Please standardize sleep disorder assessment criteria across surveys by selecting either difficulty falling asleep or awakenings (rather than using the maximum value of both) or alternatively adopting an averaged metric.
- The sample sizes of the 22 datasets vary dramatically, from 110 (JPN022HR2005) to 1,483 (JPN014CR2002). Why the manuscript does not apply weighting in regression analyses to account for this imbalance? Please implement weighted least squares regression, using sample size as weights, to mitigate bias and enhance generalizability.
- The graphical presentation of results suffers from inconsistent marker shapes and color schemes across noise sources (road traffic, conventional railway, high-speed railway). Such inconsistency complicates visual differentiation, especially for overlapping exposure ranges. Please refine the design of fitted charts to improve visual clarity and interpretability.
Author Response
Comments 1: The manuscript aggregates datasets spanning 23 years (2000–2023), which introduces potential temporal biases. Why do not address generational shifts in noise sensitivity or lifestyle changes over this period? Please conduct intergenerational comparative analyses to examine whether societal changes or cumulative noise pollution exposure influence subjective sleep disorder reporting.
Response 1: Thank you very much for your insightful comment. To examine potential changes in responses over time, we added a dummy variable in the logistic regression analysis to distinguish between surveys conducted from 2000 to 2009 and those conducted in 2010 or later. In relation to this, we have made the following revisions:
- In response to Reviewer 5's comment, the methodological descriptions previously included in Chapter 3 have been moved to Section 2.5, and we have also added an explanation of this analysis in Section 2.5 (L259-261).
- The results of the temporal trend analysis have been added to Section 3.3 (L338-342).
In Chapter 4 (Discussion, Lines 464-477), we have included a discussion on changes over time.
Comments 2: The harmonization of sleep disturbance metrics across datasets raises concerns. Surveys used divergent scales: binary (yes/no) or 5-point Likert scales evaluating difficulty falling asleep and awakenings. Why the manuscript combine responses by taking the maximum value between difficulty falling asleep and awakenings? This approach assumes equivalence between these distinct phenomena, though they may reflect different physiological and psychological impacts. Please standardize sleep disorder assessment criteria across surveys by selecting either difficulty falling asleep or awakenings (rather than using the maximum value of both) or alternatively adopting an averaged metric.
Response 2: As you pointed out, we have decided to use the average value of the scores for difficulty falling asleep and nighttime awakenings, rather than the maximum of the two. We have also included a description of the method for calculating the HSD in the manuscript (L245-249). In accordance with this analysis, we also revised the exposure–response relationships. As a result, Tables 6-9, Equations (1)–(3), and Figures 1 and 2 have been updated accordingly.
Comments 3: The sample sizes of the 22 datasets vary dramatically, from 110 (JPN022HR2005) to 1,483 (JPN014CR2002). Why the manuscript does not apply weighting in regression analyses to account for this imbalance? Please implement weighted least squares regression, using sample size as weights, to mitigate bias and enhance generalizability.
Response 3: In accordance with your suggestion, we have revised the curve estimation by applying weighting based on the number of respondents (L358-360, Equations (1)–(3), Table 9, and Figures 1 and 2).
Comments 4: The graphical presentation of results suffers from inconsistent marker shapes and color schemes across noise sources (road traffic, conventional railway, high-speed railway). Such inconsistency complicates visual differentiation, especially for overlapping exposure ranges. Please refine the design of fitted charts to improve visual clarity and interpretability.
Response 4: In response to your suggestion, we interpreted it as a recommendation to standardize the line colors in the graph by noise source, and have made the necessary revisions (Figure 2).
Reviewer 2 Report
Comments and Suggestions for Authors
The authors evaluate the existence of a relationship between noise annoyance by traffic noise and two types of railway noise in the city of Japan. The results are similar to previous results conducted by the WHO, although for raiway noise, in ares exposed to highn noise levels, the results were lower to those shown in the WHO reports. They also conduct a dissertation on the use of different scales in measuring the percentage of people highly annoyed by road traffic noise or railway noise.
The article is well-argued, uses appropriate bibliographic references, and the analysis and results have been sufficiently explained. Therefore, I consider the article acceptable for publication with the following minor modifications.
In lines 215-216, the authors indicate that “Lnight in 5-dB steps was distinguished from that in 1-dB steps, with the former abbreviated as NL in this paper”. However, I did not found the NL abbreviation before
In lines 398-399 the authors indicate that “These findings suggest that annoyance responses are strongly influenced by non-acoustic factors such as attitudes toward the noise source.” Could not be due to different acoustic isolation of the houses, socio-economical factors…? Please, go deeper on thispart of the discusión.
Author Response
Comments 1: In lines 215-216, the authors indicate that “Lnight in 5-dB steps was distinguished from that in 1-dB steps, with the former abbreviated as NL in this paper”. However, I did not found the NL abbreviation before.
Response 1: The term "DENL" is a provisional notation used to distinguish it from Lnight. In our previous study, we used the notation "DENL" to represent Lden in 5 dB increments, and we have adopted the same notation in the present study. Although this expression may not be commonly found in other previous studies, we believe it is not confusing or inappropriate in this context. Therefore, we would like to keep it unchanged.
Comments 2: In lines 398-399 the authors indicate that “These findings suggest that annoyance responses are strongly influenced by non-acoustic factors such as attitudes toward the noise source.” Could not be due to different acoustic isolation of the houses, socio-economical factors…? Please, go deeper on thispart of the discusión.
Response 2: "Attitude toward noise" refers to one's perception of the noise source, shaped by factors such as the perceived social benefits of the transportation system causing the noise and the personal advantages one may gain from it. This concept has been discussed in various studies, and we have now added relevant references to the manuscript (L432-434).
Reviewer 3 Report
Comments and Suggestions for Authors
The paper investigates the connection between noise exposure and self-reported sleep disturbance caused by traffic and railway noise. The author's findings are interesting and valuable. However, there are some limitations to the study, which the authors acknowledge. Additionally, certain aspects of the paper could be clarified to better present the results.
- The equivalent noise levels are used to describe noise exposure. But the people can be disturbed by a single pass of a train or a vehicle (for example, extremely loud bike). The peak due to noisy pass slightly influence the equivalent noise level. So, the presented analysis ignores the maximal noise levels. Is it crucial or not for the presented findings?
- There is no information available about the frequency spectra of the noise sources under consideration. It is known that the rail and vehicles have different spectral characteristics. How does the frequency of noise affect sleep disturbance?
- As we see in Table 6, the distributions of NL have maximus at different values NL. It seems that they depend on not only noise levels. What defines them? May me data collection technique (SASDA?) affect the results? If so, is it correct to compare data with different inner senses?
Author Response
Comments 1: The equivalent noise levels are used to describe noise exposure. But the people can be disturbed by a single pass of a train or a vehicle (for example, extremely loud bike). The peak due to noisy pass slightly influence the equivalent noise level. So, the presented analysis ignores the maximal noise levels. Is it crucial or not for the presented findings?
Response 1: The association between Lnight and sleep disturbance has been examined in the WHO guidelines as well as in the study by Miedema & Vos. In this manuscript, we compared our results obtained in Japan with those of international findings. Therefore, we believe that the present analysis based solely on Lnight is still meaningful. Most of the datasets registered in SASDA do not include LAmax data, which is why we were unable to conduct an analysis using LAmax in this study. We have added a note to this effect in the Introduction section (L127-131).
Comments 2: There is no information available about the frequency spectra of the noise sources under consideration. It is known that the rail and vehicles have different spectral characteristics. How does the frequency of noise affect sleep disturbance?
Response 2: The SASDA dataset does not contain information on frequency spectra, and therefore, such analyses could not be conducted. Additionally, we believe that the scientific evidence regarding the effects of noise frequency characteristics on sleep remains minimal. While we recognize this as an important issue, it is difficult to address it through secondary analysis. Accordingly, we have stated in the Introduction that, since this study is a secondary analysis based on long-term representative exposure indicators, parameters such as LAmax and frequency characteristics could not be taken into account (L127-131).
Comments 3: As we see in Table 6, the distributions of NL have maximus at different values NL. It seems that they depend on not only noise levels. What defines them? May me data collection technique (SASDA?) affect the results? If so, is it correct to compare data with different inner senses?
Response 3: One of the reasons for the wide variation in exposure levels across surveys is the differing distances from noise sources at which the social surveys were conducted. This variation stems from the fact that the locations where surveys could be implemented differed for each study. In the case of Shinkansen noise, improvements in noise mitigation measures over time may also have contributed to differences in exposure levels. Given such variability across surveys, conducting a meta-analysis that integrates these datasets allows us to examine exposure–response relationships across a broader range of exposure levels. We have added this explanation to the manuscript (L323-330).
Reviewer 4 Report
Comments and Suggestions for Authors
Nighttime noise is a problem in urban areas and those close to transport lines, but also to wind farms. The authors' contribution to the topic should be highlighted. The novelty of the paper and the research should be highlighted. Should it be explained how the datasets were chosen? With what criteria? Are the authors aware of their validity? Aircraft noise was not considered, but it is the most annoying together with wind turbines. In this article, the nighttime noise level (Lnight) is defined as the continuous equivalent A-weighted sound pressure level, during the night (from 22:00 to 07:00). But were other types of filters not considered, for example dBC? Were frequency analyses performed to evaluate low frequencies or tonal components? It should be highlighted whether the measured values ​​are from the authors of the paper or from other research. If the measurements are from other research your paper becomes very limited, because you use measurements performed by other research, a critical study of the measurements is necessary.
Il paper is not a research article, but a review paper
Author Response
Comments 1:
Nighttime noise is a problem in urban areas and those close to transport lines, but also to wind farms. The authors' contribution to the topic should be highlighted. The novelty of the paper and the research should be highlighted. Should it be explained how the datasets were chosen? With what criteria? Are the authors aware of their validity? Aircraft noise was not considered, but it is the most annoying together with wind turbines. In this article, the nighttime noise level (Lnight) is defined as the continuous equivalent A-weighted sound pressure level, during the night (from 22:00 to 07:00). But were other types of filters not considered, for example dBC? Were frequency analyses performed to evaluate low frequencies or tonal components? It should be highlighted whether the measured values ​​are from the authors of the paper or from other research. If the measurements are from other research your paper becomes very limited, because you use measurements performed by other research, a critical study of the measurements is necessary.
Il paper is not a research article, but a review paper
Response 1:
The objective of this study is to utilize data accumulated in Japan to compare the exposure–response relationship between Lnight from transportation noise and sleep disturbance with those reported in the WHO guidelines and in other countries. In Japan, the Social Survey Data Archive on Acoustics (SASDA) has compiled data from numerous social surveys on environmental noise. As a data archive of social surveys related to noise, SASDA enables meta-analyses similar to those conducted by the WHO and TNO. There have been relatively few studies that have performed meta-analyses using such a large volume of historical data from multiple studies, and we believe this research has considerable significance in that regard. We intend to clearly state this point in the introduction (L111-117).
As for data selection, we included only datasets registered in SASDA that were collected in or after the year 2000. This decision was made in consideration of changes in society and public awareness of environmental issues over time, following the criteria used by Basner & McGuire [6]. In addition, we used 11 datasets that were published after 2000 but are not yet registered in SASDA (L138-141).
With regard to specific noise sources, no datasets related to wind turbine noise are currently available in SASDA, so this source was excluded from the analysis. Similarly, only two datasets on aircraft noise are available, and in Japan, there are very few airports operating during nighttime hours (from 22:00 to 07:00). As a result, the number of respondents affected by nighttime aircraft noise was extremely limited, and these datasets were therefore excluded (L122-126).
In this study, we used the A-weighted sound pressure level, which is considered the de facto standard. Although analyses using C-weighting or frequency-specific measures might allow us to examine the potential effects of low-frequency noise, such an approach is beyond the scope of the present study. Moreover, since previous social surveys have not employed C-weighting, a meta-analysis using such data would not be feasible.
Reviewer 5 Report
Comments and Suggestions for Authors
Dear authors, I congratulate you on the development of this research, which I found very interesting.
From reading the manuscript, your observations stand out: throughout the text, the notation for sound exposure is incorrect, because instead of citing dB(A), it always mentions dB, which does not correspond to the sensation of the human ear.
I also noticed that in the results section there is a notable contamination of the applied method. For example, between lines 290 and 294, the applied logistic regression method is mentioned. It is necessary to delimit the sections, because it makes it difficult to read the results.
Author Response
Comments 1: From reading the manuscript, your observations stand out: throughout the text, the notation for sound exposure is incorrect, because instead of citing dB(A), it always mentions dB, which does not correspond to the sensation of the human ear.
Response 1: Thank you for your comments. The notation “dB (A)” is incorrect. It is not appropriate to add the symbol “(A)” to the unit “dB,” and international standards such as those issued by ISO do not use the expression “dB (A).”
It is a misconception that “dB” does not refer to A-weighted sound pressure levels. Even for A-weighted sound pressure levels, the correct unit remains “dB.”
Comments 2: I also noticed that in the results section there is a notable contamination of the applied method. For example, between lines 290 and 294, the applied logistic regression method is mentioned. It is necessary to delimit the sections, because it makes it difficult to read the results.
Response 2: Thank you very much for your valuable comment. As you pointed out, the description mixed the analytical methods with the results, making the text difficult to follow. We have revised the manuscript by moving the relevant parts, including the section you mentioned, to Chapter 2, specifically Sections 2.4 and 2.5, where the analytical methods are now clearly described.
Round 2
Reviewer 4 Report
Comments and Suggestions for Authors
The paper often reports dB, not dBA; this would be correct.
The Lden and Lnight values are reported; it should be specified how the measurements were performed.
The text row 445, the noise from hinkansen trains often does not cause disturbance at night. The reason for this should be explained.
Could you describe how the homes are built? For example, what is the soundproofing value of the windows?
The paper could be improved with noise maps.
Author Response
Comments 1: The paper often reports dB, not dBA; this would be correct. The Lden and Lnight values are reported; it should be specified how the measurements were performed.
Response 1: Thank you for your comment. Although the specific measurement methods differ slightly across datasets, all surveys except for No. 22 conducted noise measurements using sound level meters near roads or railway tracks, and estimated residential noise levels based on distance attenuation. We have added the following statement to the manuscript:
(L184-189) "Except for dataset No. 22 (KNG110RT2023), the noise levels at respondents’ residences were estimated based on field measurements. Sound level measurements were conducted at multiple points near roads or railway tracks, and noise levels at residential position were then estimated using distance attenuation. In contrast, for dataset No. 22, the estimation was based on a noise map developed using a road traffic noise prediction model."
Comments 2: The text row 445, the noise from hinkansen trains often does not cause disturbance at night. The reason for this should be explained.
Response 2: As stated in the manuscript, our intention was to point out that, due to the very limited operation of Shinkansen trains during nighttime hours, there are fewer opportunities for exposure during sleep compared to other noise sources. We did not mean to imply that the impact of Shinkansen noise itself is small. To avoid misunderstanding, we have added a brief clarification in the text. (L455-461)
Comments 3: Could you describe how the homes are built? For example, what is the soundproofing value of the windows?
Response 3: Regarding housing structure, we can clearly state that this study is limited to detached houses. However, we do not have data on the specific construction methods of individual houses, and therefore, we are unable to provide a detailed response on that point. As a general observation, most detached houses in Japan are wooden structures. Similarly, the sound insulation performance of windows varies widely, and since such data are not included in the dataset, it is difficult to address this aspect as well. Generally speaking, an open window is assumed to provide around 10 dB of sound attenuation, while a closed window can provide approximately 20 to 30 dB. As with many previous studies, this variation in sound insulation among houses represents a limitation of the current study, which relies on outdoor noise measurements. We have added this point as a limitation in the manuscript. (L521-529)
Comments 4: The paper could be improved with noise maps.
Response 4: Thank you for your comment. We agree with your point. The following statement has been added to the section on limitations.
(L530-537) "In Japan, noise policies based on noise maps—such as those commonly implemented in Europe—are still extremely limited, and the dissemination of noise maps has been slow. As a result, social surveys utilizing noise maps have rarely been conducted. Consequently, the present analysis relies on exposure estimates based on measurements conducted at limited locations, leaving issues related to uncertainty in exposure levels. Recently, however, surveys such as dataset No. 22 have begun to incorporate noise maps, and in the future, the wider implementation of noise maps is expected to enable the development of more accurate exposure–response relationships."